

# 1 Significant Improvement of Cloud Representation in Global Climate
# 2 Model MRI-ESM2

Hideaki Kawai[1], Seiji Yukimoto[1], Tsuyoshi Koshiro[1], Naga Oshima[1], Taichu Tanaka[1], Hiromasa
Yoshimura[1], Ryoji Nagasawa[1]
[1]Meteorological Research Institute, Japan Meteorological Agency, Tsukuba, 305-0052, Japan
*Correspondence to*: Hideaki Kawai (h-kawai@mri-jma.go.jp)
**Abstract.** The development of the climate model MRI-ESM2, which is planned for use in the sixth phase of the Coupled
Model Intercomparison Project (CMIP6) simulations, involved significant improvements to the representation of clouds
from the previous version MRI-CGCM3, which was used in the CMIP5 simulations. In particular, the serious lack of
reflection of solar radiation over the Southern Ocean in MRI-CGCM3 was drastically improved in MRI-ESM2. The score of
the spatial pattern of radiative fluxes at the top of the atmosphere for MRI-ESM2 is better than for any CMIP5 model. In this
paper, we set out comprehensively the various modifications related to clouds that contribute to the improved cloud
representation, and the main impacts on the climate of each modification. The modifications cover various schemes and
processes including the cloud scheme, turbulence scheme, cloud microphysics processes, interaction between cloud and
convection schemes, resolution issues, cloud radiation processes, interaction with the aerosol model, and numerics. In
addition, the new stratocumulus parameterization, which contributes considerably to increased low cloud cover and reduced
radiation bias over the Southern Ocean, and the improved cloud ice fall scheme, which alleviates the time-step dependency
of cloud ice content, are described in detail.

## 20 1 Introduction

Representation of clouds is crucially important for climate models because errors in simulated radiative fluxes are
caused mainly by poor representation of cloud rather than by errors in the clear sky radiation calculation. Consequently,
biases in clouds are the major factor for biases in the radiation budget and sea surface temperature that essentially determine
the basic performance of climate models. In addition, it is widely recognized that a large part of the uncertainty in projected
increases in surface temperature in global warming simulations by climate models arises from large uncertainties in cloud
feedback (e.g., Soden and Held, 2006; Soden et al., 2008). To obtain reliable cloud feedback in the climate models used for
the projection, clouds must be represented realistically, at least in their climatology. Therefore, cloud schemes and their
related processes are the most important atmospheric physical processes to be considered and carefully examined in the
development of climate models.




When a climate model undergoes a major upgrade with a new version name, many minor modifications are often
included rather than the introduction of a completely new sophisticated scheme. However, details are generally not provided
of such minor modifications including the technical information and the tuning of physics schemes related to clouds,
although such information is very useful and includes much scientific and technical value. Mauritsen et al. (2012) is one
example of a publication that provides practical and honest information for tuning of a climate model.
We participated in the fifth phase of the Coupled Model Intercomparison Project (CMIP5) (Taylor et al., 2012) and the
Cloud Feedback Model Intercomparison Project Phase 2 (CFMIP-2) (Bony et al., 2011) using our global climate model,
MRI-CGCM3 (Yukimoto et al., 2012, 2011). However, its representation of clouds was unsatisfactory. In the updated
version of our climate model, MRI-ESM2 (Yukimoto et al., 2019, submitted), which is planned for use in CMIP6 (Eyring et
al., 2015) and CFMIP-3 (Webb et al., 2017) simulations, the representation of clouds is significantly improved. The score of
the spatial pattern of radiative fluxes for MRI-CGCM3 was worse than the average of the 48 CMIP5 models but the score for
MRI-ESM2 is better than any of them. The improvement is particularly pronounced over the Southern Ocean. Trenberth and
Fasullo (2010) showed that a significant lack of clouds over the Southern Ocean is a serious problem in most climate models
and causes huge biases in shortwave radiative flux there. Although MRI-CGCM3 had this problem with biases that were
worse than the average CMIP5 model, the biases are dramatically reduced in the new model, MRI-ESM2.
The problems related to clouds in MRI-CGCM3 cover a broad range of issues. For instance, low cloud cover over the
mid-latitude and subtropical oceans is insufficient, the ratio of super-cooled liquid water is too small, the number
concentration of cloud droplets of the Southern Ocean clouds is inadequate, the reflection of solar radiation over the tropics
is overestimated, vertical structures of low cloud transition are unrealistic, there are several coding bugs, and ice water
content shows strong time-step dependency. To solve these problems and give a better physical basis to the processes, many
modifications were implemented in MRI-ESM2. The model update includes:
(i) the introduction of a new stratocumulus parameterization,
(ii) a modified treatment of the Wegener–Bergeron–Findeisen (WBF) process,
(iii) a modified treatment of interaction between stratocumulus and shallow convection,
(iv) an increase in the vertical resolution,
(v) the introduction of a new cloud overlap scheme,
(vi) increased horizontal resolution for the radiation calculation,
(vii) various bug fixes,
(viii) updated aerosol size distributions,
(ix) an improved cloud ice fall scheme.
Item (i) is related to the cloud and turbulence schemes, (ii) to cloud microphysics process, (iii) to interaction between the
cloud and convection schemes, (iv) and (vi) to resolution issues, (v) to cloud radiation process, (viii) to the aerosol properties,
and (ix) to numerics. Improvements and modifications in this wide range of processes contribute to the improved cloud
representation in MRI-ESM2. It is worth describing the main effect of each modification separately with the background of





the modification, and such information is very useful for model developers. We would like to emphasize again that the
improvement of climate model performance due to updates is ordinarily contributed by the cumulative effect of a lot of
modifications, some of which may seem to be minor, rather than by the introduction of a new sophisticated scheme. In this
paper, the impacts of each modification are examined by comparing the result of a control AMIP simulation using the new
model MRI-ESM2 and results of AMIP experiments in which each updated process is separately turned off.
In addition, the new stratocumulus parameterization, which contributes considerably to increased low cloud cover and
reduced radiation bias over the Southern Ocean, includes scientifically new concepts, and the improved cloud ice fall scheme,
which alleviates the time-step dependency of cloud ice content, includes technically important issues. Therefore, these two
items are described in detail in the later section.


**2 Models and experiments**

**2.1 Models**
The cloud scheme in MRI-CGCM3 (Yukimoto et al., 2012, 2011; $T_L159L48$ in the standard configuration) is a two-
moment cloud scheme developed and modified from the Tiedtke cloud scheme (Tiedtke, 1993; Jakob 2000). Cloud fraction,
cloud liquid water and cloud ice water contents (LWC and IWC), number concentrations of cloud droplets and ice crystals
are prognostic variables. The source and sink terms of cloud fraction, LWC, and IWC are calculated basically following
Tiedtke (1993): the source terms include formation of stratiform cloud due to upward motion and temperature decrease and
detrainment from convection, and sink terms include evaporation. For the temperature range from –38 to 0 ℃, deposition
nucleation is calculated based on Meyers et al. (1992), and depositional growth and evaporation for cloud ice are calculated
following Rutledge and Hobbs (1983). As processes for freezing of cloud droplets to ice crystals, immersion freezing and
condensation freezing (Bigg 1953; Murakami, 1990; Levkov et al., 1992; Lohmann, 2002), and contact freezing (Lohmann
and Diehl, 2006; Cotton et al., 1986) are calculated. Conversion of LWC to rain is calculated based on Manton and Cotton
(1977) and Rotstayn (2000). Melting of cloud ice and snow occurs just below an altitude where the atmospheric temperature
is 273.15 K. In MRI-ESM2 (Yukimoto et al. 2019, submitted; $T_L159L80$ in the standard configuration), all these processes
are essentially the same as in MRI-CGCM3. The treatments of stratocumulus, the Bergeron–Findeisen effect, cloud ice fall,
and conversion of IWC to snow are discussed later in detail because they are modified from MRI-CGCM3 to MRI-ESM2.
Aerosols are calculated by the Model of Aerosol Species in the Global Atmosphere mark-2 revision 4-climate
(MASINGAR mk-2r4c) (Yukimoto et al., 2011; Tanaka et al., 2003; Yukimoto et al., 2019, submitted), which is coupled to
MRI-ESM2. Five species of aerosols are utilized in the cloud and radiation schemes: sulfate, black carbon, organic matter,



sea salt (2 size modes), and mineral dust (6 size bins). The activation of aerosols into cloud droplets is calculated based on
Abdul-Razzak et al. (1998), Abdul-Razzak and Ghan (2000), and Takemura et al. (2005). The ice nucleation for cirrus
clouds is calculated using a parameterization of Kärcher et al. (2006), including homogeneous nucleation (Kärcher and
Lohmann, 2002) and heterogeneous nucleation (Kärcher and Lohmann, 2003).

## 2.2 Basic performance

First, we briefly show improvements from MRI-CGCM3 to MRI-ESM2 in the basic performance of the simulations.
Figure 1 shows the total cloud cover and its bias in the present-day climate from the historical simulations using MRI-
CGCM3 and MRI-ESM2. Observational data for total cloud cover (Pincus et al., 2012; Zhang et al., 2012) that are derived
from the International Satellite Cloud Climatology Project (ISCCP; Rossow and Schiffer, 1999) D1 data and radiative flux
observational data from the Clouds and Earth's Radiant Energy Systems (CERES) Energy Balanced and Filled (EBAF; Loeb
et al., 2009) product are used as observational climatologies. It is clear that total cloud cover simulated by MRI-CGCM3 is
much less than the observations, especially over the Southern Ocean and subtropical oceans off the west coast of the
continents. However, total cloud cover is substantially increased in the simulation using MRI-ESM2 over these areas and the
bias is reduced significantly. As a result, a large negative bias in the upward shortwave radiative flux at the top of the
atmosphere (TOA) found in MRI-CGCM3 is reduced substantially in the simulation using MRI-ESM2. In addition, a
positive bias in the tropics is also reduced.
Figure 2 shows the Taylor diagrams (Taylor, 2001) for upward shortwave, longwave, and net radiative fluxes from the
48 CMIP5 models. The scores of spatial patterns of shortwave, longwave, and net radiative fluxes for MRI-CGCM3 are near
or worse than the average among the 48 CMIP5 models, but the scores for MRI-ESM2 are better than any of the models. The
scores for MRI-ESM2 are even almost comparable to the scores of the ensemble mean of CMIP5 models. Although the
uncertainty in the observational data for cloud radiative effect is larger than that of radiative fluxes at the top of the
atmosphere, the scores of cloud radiative effect for shortwave, longwave, and net radiation show similar characteristics to the
corresponding scores for TOA radiative fluxes (Fig. S1). This implies that improvement of TOA radiative fluxes in MRI-
ESM2 can be attributed to improvement of cloud representation in the model.

## 2.3 Experiments

The purpose of this paper is to identify the effect of each modification applied to the model under controlled conditions
in order to understand the significant improvement of the radiative flux in the new model. Therefore, we chose AMIP
simulations to avoid being influenced by changes in SST. A series of experiments with the new model MRI-ESM2 is
performed, with each modification summarized in Sect. 1 in turn set to the old (MRI-CGCM3) treatment. A list of sensitivity



experiments performed in the present study using MRI-ESM2 is given in Table 1. We ran the model from 2000 to 2010 and
used the data for 10 years from 2001 to 2010 for analysis.


## 3 Updates and their impacts

In this section, the updates from various aspects are explained with their backgrounds. The main impact of each update
is shown and discussed based on the comparison between the results of the updated new model and the experiments in which
each modification in turn is turned back to the old treatment.

### 3.1 New stratocumulus parameterization

Representation of low clouds including stratocumulus in climate models has been one of the most bothersome problems
for many years (e.g., Duynkerke and Teixeira, 2001; Siebesma et al., 2004), and low clouds are poorly reproduced even in
the state-of-the-art climate models (e.g., Nam et al., 2012; Su et al., 2013; Caldwell et al., 2013; Koshiro et al., 2018). As a
result, solar reflectance by clouds has significant negative biases over areas frequently covered by stratocumulus (e.g.,
Trenberth and Fasullo, 2010; Li et al., 2013). A new stratocumulus scheme that utilizes a stability index that takes into
account the effect of cloud top entrainment (Kawai et al., 2017) was introduced instead of the old stratocumulus scheme
(Kawai and Inoue, 2006). A detailed description and physical interpretation are given in Sect. 4. Figure 3 shows that low
cloud cover increases significantly in the subtropical oceans off the west coast of the continents and over the Southern Ocean,
which is a significant result of upgrading the stratocumulus scheme. Low cloud cover is increased by more than 20% over
the oceans off California, Peru, Namibia, and west coast of Australia, and by more than 10% over the Southern Ocean. As a
result, upward shortwave radiative flux (reflection of solar insolation) also increases and this impact contributes to reducing
the large bias in shortwave radiative flux over these regions.

### 3.2 Treatment of the WBF effect

In recent years, several studies (e.g., McCoy et al., 2015; Cesana and Chepfer, 2013) revealed that ratios of super-
cooled liquid water with respect to cloud (liquid + ice) water in climate models are much lower than those in the Cloud–
Aerosol Lidar and Infrared Pathfinder Satellite Observations (CALIPSO; Winker et al., 2009) data (e.g., Hu et al., 2010;
Cesana and Chepfer, 2013). Some studies pointed out that the lack of super-cooled liquid water in climate models is the
source of insufficient solar reflectance of clouds over the Southern Ocean (e.g., Bodas-Salcedo et al., 2016; Kay et al., 2016).
Liquid clouds are optically thicker than ice clouds if the cloud (liquid + ice) water content is the same, because the size of



cloud droplets is much smaller than that of ice crystals and this corresponds to larger number concentration for cloud
droplets.
The WBF process is a deposition growth process of ice crystals at the expense of cloud droplets due to ice saturation
being lower than liquid water saturation. The WBF effect was treated in a way similar to Lohmann et al. (2007) in MRI-
CGCM3. When IWC is greater than a threshold of 0.5 mg kg$^{-1}$, all super-cooled water in the grid box is forced to evaporate
within the time step and all source terms for LWC are set to zero. However, this treatment caused excessive evaporation of
super-cooled water. In MRI-ESM2, when IWC exceeds the threshold, only the part of LWC that corresponds to the
depositional growth of ice crystals is evaporated within the time step. In addition, the source terms of LWC are not ignored
but calculated in a proper fashion. However, there is an arbitrariness about how these source terms are divided into the
source terms of LWC and IWC. The first reason for the arbitrariness is that the time step of our climate models is too long
(30 minutes) to resolve cloud microphysics and a part of the generated liquid water can change to ice crystals within this
time step, especially when IWC exceeds the threshold. The second reason is that the liquid water and ice water are assumed
to be well mixed in the model grid box if they coexist, as in most global climate models. However, there should be mixed
phase parts, ice only parts, and liquid only parts in a volume corresponding to the model grid box size (Tan and Storelvmo
2016). Therefore, it is difficult to determine the LWC–IWC partitioning of the source terms theoretically. We decided to use
a ratio derived by Hu et al. (2010) based on satellite observations to determine the ratio of the source terms into LWC and
IWC only when the WBF effect occurs, that is, when IWC is greater than the threshold. This is an empirical and simple
method, but this treatment can supplement the defects of the modelled microphysics due to the uncertainty and complexity
by utilizing observational data.
Figure 4 shows the ratio of super-cooled liquid water in clouds as a function of temperature in the simulations using
new and old treatments of the WBF effect. It is clear from the figure that the ratio of super-cooled liquid water is
significantly increased in the new treatment and close to the satellite observations of Hu et al. (2010); the ratio at 255 K is
increased from 52% to 84% for the mass-weighted ratio and from 18% to 78% for the frequency ratio. Both mass-weighted
ratio and frequency ratio, which should correspond to the ratio derived from satellite observations, using the new treatment
are close to the satellite observations. Figure 5 shows the impact of the new treatment of the WBF effect on TOA upward
shortwave radiative flux. The reflection of solar insolation is significantly increased over the Southern Ocean using the new
treatment (Fig. 5), and consequently, this new treatment contributes considerably to the reduction in shortwave radiation bias
over the area shown in Fig. 1. The increase in the ratio of super-cooled liquid water in MRI-ESM2 plausibly contributes to
the higher climate sensitivity in the model than in MRI-CGCM3, because an increased ratio of super-cooled liquid water
weakens the cloud-phase feedback that negatively contributes to cloud feedback (Tsushima et al., 2006; McCoy et al., 2015;
Bodas-Salcedo et al., 2016; Kay et al., 2016; Tan et al., 2016; Frey and Kay, 2018).
However, since the new treatment of the WBF effect is still rather simple, it cannot represent observed layered
structures with a thin super-cooled water layer at the top of cloud layers and ice layer below (Forbes and Ahlgrimm, 2014;
Forbes et al., 2016). In addition, it is possible that the curve of Hu et al. (2010) over-estimates the ratio of super-cooled



liquid water (Cesana and Chepfer, 2013; Cesana et al., 2016). Therefore, more sophisticated treatments need to be developed
in the future.

**3.3 Interaction between stratocumulus and shallow convection**
It is well-known that the altitude of the low-level cloud layer gradually increases westward in subtropical stratocumulus
regions, including off Peru, in association with the transition from stratocumulus to cumulus (Bretherton et al., 2010; Rahn
and Garreaud, 2010; Abel et al., 2010; Kawai et al., 2015). However, the vertical structures of the transition were
unrealistically discontinuous in the old model as seen in Fig. 6b. This discontinuity was caused by an unrealistically formed
temperature inversion just above the stratocumulus-like cloud layer due to excessive adiabatic heating by the convection
scheme that activates shallow convection in those regions. Therefore, in the new version, the occurrence of shallow
convection is prevented over the area where the conditions for stratocumulus occurrence (See Section 4.1 in more detail) are
met. As a result, the vertical structures of low-level clouds are significantly improved, as seen in Fig. 6a. Such a switch for
shallow convection is sometimes used in atmospheric models, although it is a simple and practical method. For example, a
threshold of estimated inversion strength (EIS; Wood and Bretherton, 2006) is used to determine the activation of shallow
convection in version CY43r3 of the European Centre for Medium-Range Weather Forecasts (ECMWF) Integrated Forecast
System (IFS) (ECMWF, 2017).

**3.4 Vertical resolution**
The thickness of observed stratocumulus is typically 200−300 m (Wood 2012), but can be as thin as 50 m during the
daytime, especially in the Californian stratocumulus region (Betts, 1990; Duynkerke and Teixeira, 2001). The model vertical
resolution was increased from L48 (48 vertical levels) in the MRI-CGCM3 to L80 in the MRI-ESM2 (Yukimoto et al. 2019,
submitted), and the number of vertical layers in the atmospheric boundary layer was nearly doubled (from 5 to 10 layers
below 900 hPa). As seen in Fig. 6c, the low cloud layer can be geometrically too thick in the model with resolution L48,
which can cause too high an albedo, because the vertical layer thickness is about 300 m at the level of 900 hPa and this is the
minimum thickness of clouds that can be represented in the model. Sensitivity of represented stratocumulus to model vertical
resolution has been widely reported (Teixeira, 1999; Bushell and Martin, 1999; Wang et al., 2004; Wilson et al., 2008;
Neubauer et al., 2014; Guo et al., 2015). Although several methods that compensate for insufficient vertical resolution have
been developed, including the use of vertical sub-levels (Wilson et al. 2007) and the introduction of areal cloud fraction,
which is different from volume cloud fraction (Brooks et al., 2005), we decided for the moment not to introduce those
methods for simplicity and consistency in the model physics.





**3.5 Cloud overlap**

In the longwave radiation scheme, maximum-random overlap (Geleyn and Hollingsworth, 1979) is adopted as a cloud overlap assumption. In contrast, in the shortwave radiation scheme, total cloud cover in a column (the cloudy area) is first calculated based on maximum-random overlap, and second, random overlap is adopted indirectly to calculate multiple scattering in the cloudy area in the MRI-CGCM3 (Yukimoto et al., 2011, 2012). However, the inadequate treatment of the cloud overlap assumption in the shortwave radiation scheme causes overestimation of the reflection of incident solar radiative flux, especially for tower-shaped cumulus clouds with optically thin high-level clouds (e.g. anvil) (Nagasawa, 2012). In MRI-ESM2, because a practical independent column approximation (PICA; Nagasawa, 2012) based on Collins (2001) was implemented, the maximum-random overlap became available in the shortwave radiation scheme. Application of the maximum-random overlap in the shortwave radiation scheme significantly decreased the reflection of shortwave radiative flux over the tropical convection areas without varying total cloud cover (Fig. 7). This reduction makes a significant contribution to reduce the excessive reflection of incident shortwave radiative flux over the tropics (see Fig. 1).

**3.6 Horizontal resolution for radiation calculation**

The computational cost for radiation calculation is heavy in climate models and this cost was reduced in MRI-CGCM3 by reducing the radiation calculation spatially and temporally. Full radiation computations were performed for every two grid boxes in the zonal direction, and shortwave and longwave radiation was calculated 1-hourly and 3-hourly, respectively. Figure 8 shows the impacts of increased horizontal resolution for the radiation calculation (calculation for every single grid) (Fig. 8a, 8b) and increased frequency of calculation (1-hourly calculation) for longwave radiation (Fig. 8c, 8d). In both cases, low-level clouds in the subtropics off the west coasts of the continents and at mid-latitudes increased, increasing shortwave reflectance a little. This increase in low cloud cover can be attributed to improved cloud–radiation interactions: cloud-top longwave cooling of low clouds, which is the primary physical process to maintain low clouds (e.g., Wood 2012), is consistently calculated at the top of existing low clouds without spatial smoothing and temporal inconsistency. Either modification is physically appropriate and improves the representation of low clouds. However, the total computational cost was increased by 5% for the spatial resolution modification and by 10% for the temporal resolution modification. Considering cost and merit comprehensively, we decided to adopt the modification only for the spatial resolution and keep the temporal treatment unchanged.

**3.7 Bug fixes**

No climate models are free from coding bugs, and they sometimes exert significant impacts on model results, although they are rarely documented in publications. MRI-CGCM3 also had some bugs that affect the simulation results to some



extent. One of them is associated with the prognostic equations for number concentrations of the cloud particles. This bug
caused a problem of large number concentrations of cloud particles leading to excessive optical thickness and accompanying
excessive reflection of solar radiation, particularly for stratocumulus and stratus over the subtropics and northern Pacific
region (Tsushima et al., 2016). In addition, the bug caused a large decrease in the number concentration of cloud droplets
and large positive cloud feedback for such clouds in warmer climate simulations (Kawai et al. 2015). Several bugs including
this serious bug were fixed in MRI-ESM2.

## 258  3.8 Aerosol size distributions

Our climate models calculate number concentrations of aerosols from the mass concentrations using the prescribed
aerosol size distributions, and the number concentrations are used to calculate number concentrations of cloud particles.
Therefore, an appropriate treatment of the aerosol size distributions is important to estimate the aerosol effect on clouds.
Aerosol size distributions, namely the geometric mean radius and standard deviation in lognormal size distribution, were
modified in MRI-ESM2 based on recent observations. For example, the increase in the geometric mean radius of organic
carbon from 0.0212 (Chin et al., 2002) to 0.1 μm (Seinfeld and Pandis, 2006; Liu et al., 2012) in MRI-ESM2 causes a
significant decrease in the number concentration of cloud particles that originate from organic carbon. This modification
significantly decreases the response of cloud optical thickness to assumed changes in the emission of organic carbon. On the
other hand, the mode radius of fine mode sea salt is decreased from 0.228 (Chin et al., 2002) to 0.13 μm (Seinfeld and Pandis,
2006) and the change causes higher number concentration of cloud droplets originating from sea salt. In addition, the number
concentration of cloud condensation nuclei (CCN) originating from fine mode sea salt is multiplied by a factor of 2.0 after
the calculation from the number concentration of sea salt. This treatment is introduced because we use only two size modes
(i.e., fine accumulation and coarse modes) of sea salt and the model cannot represent sea salt in the Aitken mode, although a
part of the sea salt in Aitken mode can work as CCN. Actually, the number concentration of sea salt in Aitken mode is
difficult to estimate from the mass concentration of aerosols because they contribute substantially to the number but
contribute little to the mass. To represent the contribution of sea salt in Aitken mode to CCN in a simple way, the factor of
2.0 is multiplied as a provisional solution until sea salt in Aitken mode can be calculated explicitly. This factor is estimated
from observational studies (e.g., Covert et al., 1996; Clarke et al., 2006). In fact, a lower limit of the number concentration of
cloud droplets has been used in a significant number of state-of-the-art climate models to prevent too small number
concentrations of cloud droplets in clean air conditions (Hoose et al., 2009; Jones et al., 2001; Lohmann et al., 2007;
Takemura et al., 2005). However, it is pointed out that this lower limit drastically controls the magnitude of the aerosol
indirect effect, for instance, measured as the difference between present-day and preindustrial climates (Hoose et al., 2009).
Therefore, the lower limit of cloud droplets is not introduced in our model. We believe that our treatment is better than
introducing a lower limit of cloud droplets although it is quite simple, because the treatment has a more physical basis. This





treatment increases cloud droplet number concentration by more than 30% and also increases reflection of shortwave
radiation by 4 W m$^{-2}$ over the Southern Ocean (Fig. 9).

**3.9 Ice sedimentation and ice conversion to snow**
The method for calculating cloud ice sedimentation in the MRI-CGCM3 was not sophisticated, and it caused unrealistic
ice sedimentation and strong time-step dependency of IWC. While IWC is a prognostic variable in the MRI-CGCM3, snow
is not but it is treated as snow flux in the model. A part of IWC is diagnosed as snow and removed from the IWC at each
time step and falls down to the surface within one time step. The main problem was that the ratio of snow was not
proportional to the time step. As a result, a substantial amount of snow is repeatedly removed from IWC when the time step
is shortened. To solve the problem, the treatment of cloud ice sedimentation and conversion of cloud ice to snow was
improved based on the study of Kawai (2005). Figure 10 shows that IWC is large for a time step of 3600 s but monotonically
decreases with shorter time steps. On the other hand, IWC is not affected by the time step in the control simulation that uses
the modified scheme of ice sedimentation and ice conversion to snow. A detailed description of the modification is given in
Sect. 4, because this modification contains some important insights and solutions related to the numerical issues.

**3.10 Summary of impacts on shortwave radiative flux**
Figure 11 summarizes the impacts of each modification on zonal means of low cloud cover and TOA upward
shortwave radiative flux. The new stratocumulus scheme contributes to an increase in low cloud cover mainly over the
Southern Ocean, and the suppression of shallow convection under stratocumulus conditions contributes a low cloud cover
increase over the mid-latitudes in the Southern Hemisphere. Increased horizontal resolution in the radiation calculation
additionally contributes to the low cloud cover increase. The increase in reflection of solar radiation over the Southern Ocean
and mid-latitudes in the Southern Hemisphere is largely contributed by the new stratocumulus scheme, the new treatment of
the WBF effect (especially around 60°S), the doubled number concentration of sea salt CCN, and the treatment of shallow
convection suppressed under stratocumulus conditions (over latitudes lower than the areas impacted by other modifications).
The new treatment of the WBF effect and doubled number concentration of sea salt CCN increase the reflection of solar
radiation by increasing cloud optical thickness. A new cloud overlap scheme, PICA, contributes to reduction in solar
radiation reflection over the tropics without changing the cloud cover. These modifications in MRI-ESM2 significantly
reduce the large bias in the solar radiation reflection present in MRI-CGCM3, which is negative over the Southern Ocean
and positive over the tropics (Fig. 1e, 1f, and Fig. 11c). Note that the significant improvement in the shortwave radiative flux
is not attributed to the introduction of a new advanced scheme but to the cumulative effect of many minor modifications.






## 4 Detailed description of schemes

In this section, modifications and improvements in two schemes are explained in detail, because they include scientifically new concepts and technically important insights and solutions related to the numerical issues; one is the new stratocumulus parameterization and the other is the improved cloud ice fall scheme.

### 4.1 New stratocumulus parameterization

#### 4.1.1 Old parameterization and problems

In the MRI-CGCM3, a stratocumulus scheme slightly modified from Kawai and Inoue (2006), originally developed from Slingo (1980, 1987), was used to represent subtropical stratocumulus. In that scheme, stratocumulus is formed when the following four conditions are met: (i) there is a strong inversion above the model layer, (ii) the layer near the surface is not stable (to guarantee existence of a mixed layer), (iii) the model layer height is below the level of 940 hPa, and (iv) the relative humidity of the model layer exceeds 80%. When all of these conditions are met, cloud cover is determined as a function of the inversion strength, in-cloud cloud water content is determined to be proportional to the saturation specific humidity, and the vertical mixing at the top of the cloud layer is reduced to approximately zero to prevent excess cloud top entrainment.

Although this scheme can reproduce subtropical stratocumulus and the cloud radiative effect relatively well, it has several problems. First, it does not give enough low clouds over mid-latitude oceans, especially the Southern Ocean. Low clouds off the west coast of the continents, including off California, off Peru, and off Namibia, are also insufficient, especially areas far from the coast. The second problem is related to the use of inversion strength in parameterization in climate models, which is calculated from the difference of potential temperature between two adjacent vertical model layers. Climate models cannot reproduce realistic strong inversions because their vertical resolution is totally insufficient. Furthermore, the inversion strength reproduced in climate models strongly depends on the model vertical resolution. Therefore, the parameter has to be tuned for each model, if the inversion strength is directly utilized in the parameterization. In addition, there is a strong positive feedback between cloud fraction of low cloud and the inversion strength at the top of the cloud. The positive feedback makes it difficult to utilize inversion strength in the parameterization of low cloud fraction. The third problem is that the vertical structure with a smooth transition from stratocumulus to cumulus cannot be reproduced because the parameterization is limited to below the level of 940 hPa (see Kawai and Inoue, 2006). To solve these problems, we decided to utilize a criterion that represents the structure of the lower troposphere as a whole ("non-local") rather than a detailed local vertical structure.

#### 4.1.2 New index for low cloud cover

Estimated inversion strength (EIS; Wood and Bretherton, 2006), which is a modification of lower tropospheric stability (LTS; Klein and Hartmann 1993), is an index that correlates well with low cloud cover and has been used in many studies.





However, EIS takes into account only the temperature profile and does not include information on water vapour. Kawai et
al. (2017) developed an index for low cloud cover, the estimated cloud-top entrainment index (ECTEI). This index is
deduced from a criterion of cloud top entrainment (Randall, 1980; Deardorff, 1980; Kuo and Schubert, 1988; Betts and
Boers, 1990; MacVean and Mason, 1990; MacVean, 1993; Yamaguchi and Randall, 2008; Lock, 2009) and includes
information on both the vertical profile of temperature and that of water vapour. The definition of ECTEI is as follows:
$$\text{ECTEI} \equiv \text{EIS} - \beta L / c_p (q_{\text{surf}} - q_{700})$$
where $L$ is latent heat, $c_p$ is the specific heat at constant pressure, $q_{\text{surf}}$ and $q_{700}$ are the specific humidity at the surface and
700 hPa, respectively, $\beta = (1 - k) \, C_{\text{qgap}}$, $C_{\text{qgap}}$ is a coefficient (= 0.76), and $k$ is a constant (= 0.70; MacVean and Mason

1990).

Figure 12 shows the climatologies of low stratiform cloud cover and the stability indexes, LTS, EIS, and ECTEI, for
December to February and June to August. Cloud cover data were obtained from shipboard observations, the extended edited
cloud report archive (EECRA; Hahn and Warren, 2009), and stability indexes were calculated using the ECMWF 40-year
Re-Analysis (ERA-40) data (Uppala et al., 2005) for 1957–2002. The definition of low cloud cover (LCC) in the
observations is the combined cloud cover of stratocumulus, stratus, and sky-obscuring fog, which is the same conventional
definition as employed in Klein and Hartmann (1993) and Wood and Bretherton (2006). When LCC and LTS maps are
compared, the contrast between the subtropics and mid-latitudes is different. LTS is weighted more over the subtropics than
over mid-latitudes while LCC is dominant over mid-latitudes. In EIS maps, the value is more weighted in mid-latitudes than
in the subtropics, compared with LTS, and the EIS geographical patterns are closer to LCC patterns than LTS patterns, as it
is well-known that EIS corresponds to LCC better than LTS. In ECTEI maps, the weight is even larger in mid-latitudes than
for EIS and the ECTEI geographical patterns are even closer to LCC patterns than the EIS patterns. These characteristics
suggest that EIS does not adequately represent the large occurrence of low cloud over cold oceans including the Southern
Ocean and ECTEI can be more appropriate for representation of LCC. Figure 13 shows the relationships between the LCC
and the stability indexes, LTS, EIS, and ECTEI. It shows that ECTEI has the best correlation with LCC with correlation
coefficients $R = 0.23$ for LTS, $R = 0.83$ for EIS, and $R = 0.90$ for ECTEI.
**4.1.3 New parameterization and improvements**
In our new scheme, the relationship between ECTEI and LCC is not directly used but ECTEI is used as a threshold of a
treatment in the turbulence scheme. In our climate models, vertical smoothing of vertical diffusivity is employed to represent
simply the mixing effect due to cloud top entrainment and part of the mixing due to shallow convection. In MRI-ESM2, if
ECTEI is larger than a threshold value, the smoothing is prevented, which means the turbulence at the top of the boundary
layer is suppressed, and the lower limit of vertical diffusivity is set to a much smaller value (virtually zero) than the original
one. This means that cloud top entrainment in the model is switched on and off depending on an ECTEI threshold. In the
original setting, the threshold value was set to 0 K and the condition of not stable near-surface layer (to guarantee existence
of a mixed layer) was imposed (Kawai 2013). However, after model tuning, the threshold value of ECTEI was set to −2.0 K,



and the condition of mixed layer existence was removed to apply the suppression of cloud top mixing not only to
stratocumulus conditions but also to advection fog conditions, where the near-surface layer is stable. The introduction of this
scheme has led to an increase in low cloud cover, especially over the mid-latitude ocean, including the Southern Ocean, and
the radiation bias is significantly reduced (Fig. 3).
The application of a condition that represents the detailed local vertical structure may appear to be more physically
based than a "non-local" condition. However, parameterizations based on local vertical structures are not appropriate in some
cases where (i) model resolution is not sufficient to represent the detailed physical process or (ii) the feedback between the
parameters and the variables that should be obtained is very strong. In such cases, the parameters that represent the whole
structure of the lower troposphere can produce more robust and reasonable results, although empirical relations are required
to construct "non-local" parameterizations.

## 4.2 Ice sedimentation and ice conversion to snow

### 4.2.1 Old treatment and problems

Treatment of ice sedimentation in climate models is awkward because the product of the terminal velocity of cloud ice
$v_{ice}$ (typical value ~ 0.5 m s$^{-1}$) and the time step $\Delta t$ (for example, 1800 s in MRI-CGCM3 and MRI-ESM2) can exceed the
thickness of the vertical layer $\Delta z$ (~ 500 m) in climate models. In such cases the explicit calculation is invalid and numerical
instability may occur because a vertical Courant–Friedrichs–Lewy (CFL) condition is violated. To avoid this problem,
various measures have been taken. Rotstayn (1997) reviewed the following four treatments: (A) to set an artificial limit to
the sedimentation flux for preventing defective calculation; (B) to adopt a 'fall-through' assumption; (C) to use an implicit
scheme; and (D) to use an analytically integrated scheme. Discussing the problems associated with each treatment, he
concluded that the last one (D) was the most suitable.
In MRI-CGCM3, IWC was divided into ice crystals and snow using a size threshold of 100 μm. The size distribution of
ice particles is assumed to follow a Marshall–Palmer distribution as described in Rotstayn (1997):
$$P_i(D_i) = \lambda_i e^{-\lambda_i D_i}$$

where $D_i$ is the diameter of ice particles, $\lambda_i$ is the slope factor, and the distribution $P_i(D_i)$ is normalized to 1. The slope factor
can be written as follows:
$$\lambda_i = \left(\frac{\pi \rho_i N_i}{\rho_a q_i}\right)^{1/3}$$

where $\rho_i$ is the density of ice, $N_i$ is the number concentration of ice crystals, $\rho_a$ is air density, and $q_i$ is IWC. The ratios of
cloud ice crystals with size less than 100 μm with respect to total ice crystals can be obtained analytically by integrating the
probability density function as follows:

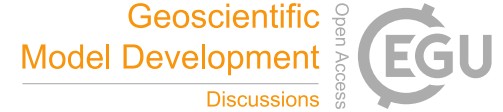

$$r_{iw} = 1 - \frac{1}{6}\{(\lambda_i D_{100})^3 + 3(\lambda_i D_{100})^2 + 6(\lambda_i D_{100}) + 6\}e^{-\lambda_i D_{100}}$$

$$r_{in} = 1 - e^{-\lambda_i D_{100}}$$

where $D_{100}$ is particle size of 100 μm, and $r_{iw}$ and $r_{in}$ are ratios of cloud ice crystals for mass and number concentrations. A
sedimentation velocity is calculated based on Heymsfield (1977), Heymsfield and Donner (1990), and Rotstayn (1997):
$$v_{\mathrm{ice}} = 3.23\left(\frac{\rho_a q_i r_{iw}}{a}\right)^{0.17} \tag{1}$$
where $a$ is cloud fraction. Ice crystals of $r_{iw}\, q_i$ fall with sedimentation velocity $v_{\mathrm{ice}}$, and snow mass $(1 - r_{iw})\, q_i$ is assumed to
fall down to the surface within a time step. Removal of the snow part based on this kind of diagnostic partition is used in
some cloud schemes. In version CY25r1 of the ECMWF IFS (ECMWF 2002), IWC is divided into two categories with sizes
larger and smaller than 100 μm following a function in McFarquhar and Heymsfield (1997; hereafter, MH97) and the larger
size portion of IWC is considered to fall through to the ground within a time step. In MRI-CGCM3, the equation of IWC to
be solved is as follows:
$$\frac{\partial q_i}{\partial t} = C_g + \frac{1}{\rho_a}\frac{\partial}{\partial z}(v_{\mathrm{ice}}\rho_a r_{iw}q_i) - \frac{(1-r_{iw})q_i}{\Delta t} \tag{2}$$
where $C_g$ is the generation rate of IWC and $\Delta t$ is the model time step. An analytically integrated solution (Rotstayn 1997;
ECMWF 2002) was used to obtain IWC after one time step.
However, this treatment contains some problems. The first is that a part of cloud ice larger than 100 μm is eliminated
from the atmosphere repeatedly when a short time step is used, because the shape of the size distribution and the ratio of ice
portions larger than and smaller than 100 μm is insensitive to IWC change. This causes strong time-step dependency of
IWC: IWC monotonically decreases with shorter time steps from 3600 s to 300 s as seen in Fig. 10. The second problem is
that the sedimentation velocity calculated from Eq. (1) is too large for ice with size smaller than 100 μm. This is because the
sedimentation velocity is supposed to represent a weighted value for the whole ice content that includes all sizes of ice, and
sedimentation velocity varies widely with particle size.

**4.2.2 New scheme and improvements**

Considering the wide range of sedimentation velocity, the velocities of falling cloud ice representing both small and
large particles are derived separately (originally reported in a preliminary report, Kawai 2005). Observed size-distribution
functions of cloud ice of MH97 and size–velocity relationships for cloud ice (Heymsfield and Iaquinta 2000) were integrated
over size using a procedure similar to Zurovac-Jevtić and Zhang (2003). See Supplement A for the detailed derivation. While
they derived only one velocity representing the total cloud ice, two velocities are derived in this study for a more
sophisticated treatment of sedimentation. The ice-fall velocity for particles smaller [larger] than 100 μm, $v_i$ [$v_s$] (m s$^{-1}$), is
obtained as a function of ice water content smaller [larger] than 100 μm, IWC$_{<100}$ [IWC$_{>100}$] (kg m$^{-3}$), as below (note that the
unit is not (kg kg$^{-1}$) but (kg m$^{-3}$)):



$\qquad v_i = 1.56(\text{IWC}_{<100})^{0.24}$ (3)
$\qquad v_s = 2.23(\text{IWC}_{>100})^{0.074}$ (4)
Figure 14 shows the velocities $v_i$ and $v_s$. The velocity of cloud ice smaller than 100 μm is much smaller than the
conventionally used velocity of ice of Rotstayn (1997). Therefore, it is inappropriate to represent the velocity of ice with size
smaller than 100 μm using the velocity of Eq. (1), and Eq. (3) is more appropriate for calculating the velocity. The figure
also shows that cloud ice larger than 100 μm has a velocity of about 1 m s$^{-1}$. Therefore, the sedimentation cannot be
calculated appropriately with the time step used in climate models, and the treatment of instant fall of snow (large ice)
through to the surface is unavoidable.
In MRI-CGCM3, it was assumed that the ratio of snow calculated from the Marshall−Palmer distribution can be
applied anytime and anywhere without taking account of the history of the cloud processes. In this case, conversion of ice
crystal into snow is not proportional to model time step and it causes the strong time-step dependency of IWC. If a
conversion rate of ice crystals into snow is available, we can avoid this time-step dependency. To obtain the rate, we assume
that the ratio given by MH97 may be regarded as a ratio between ice crystals and accumulated snow from the layers above,
which is converted from ice crystals at a certain rate. In this concept, the ratio of snow should increase as the depth from the
cloud top increases. In the derivation of the rate $C_{\text{I2S}}$ (kg kg$^{-1}$ s$^{-1}$), simple assumptions were introduced: (a) the concentration
of cloud ice is vertically homogeneous, (b) produced snow concentration is accumulated downward, (c) the observation
depth of the ratio is $H_c$ (m) from the top of a cloud. Under these assumptions, the rate can be obtained as follows (see
Appendix A for the derivation):
$\qquad C_{\text{I2S}} = \dfrac{1-\alpha_i}{\alpha_i}\dfrac{v_s}{H_c}q_i$ (5)
where $\alpha_i$ is the ratio of cloud ice content with particle sizes smaller than 100 μm to the total cloud ice content (see
Supplement A.2 for details: Fig. S2 shows $\alpha_i$ and the equation is Eq. (S10)). In this study, $H_c$ =2,000 m is assumed in
reference to MH97. The equation of IWC to be solved is as follows:
$\qquad \dfrac{\partial q_i}{\partial t} = C_g + \dfrac{1}{\rho_a}\dfrac{\partial}{\partial z}(v_i \rho_a q_i) - D_{\text{I2S}}q_i$ (6)
where $D_{\text{I2S}} = C_{\text{I2S}}/q_i$. An analytically integrated solution is used to obtain IWC after one time step.
Figure 10 shows that IWC is not affected by time step in the control simulation that uses the modified scheme of ice
sedimentation and ice conversion to snow, while the old scheme that was used in MRI-CGCM3 shows strong time-step
dependency. The improvement can mainly be attributed to the fact that the conversion of ice to snow is proportional to the
time step: the last term of the right-hand side in Eq. (6) does not explicitly depend on $\Delta t$, while the one in Eq. (2) does. In
addition, the slower sedimentation velocity in the new formulation contributes to more reasonable calculation of ice crystal
sedimentation because processes with short time-scales compared to the model time step may be unphysically calculated. In
many climate models, the terminal velocity of cloud ice has been represented by a single velocity whose typical value is ~0.5
m s$^{-1}$ (e.g., Heymsfield 1977, Heymsfield and Donner 1990), and the whole cloud ice content in the grid box falls with that



velocity (e.g., Rotstayn 1997; Smith, 1990). However, as is evident from Fig. 14, the velocity of ice crystals smaller than 100
μm is ~0.1 m s$^{-1}$ and much smaller than the typical value representing all sizes (~1 m s$^{-1}$). Small size ice crystals should
remain in the air for longer. On the other hand, some models diagnose the removal of snow portion from the total IWC
assuming a fixed size distribution without taking the history of the cloud processes into account (e.g., ECMWF 2002).
However, this causes time-step dependency, as discussed above. Note also that size distribution must change depending on
the distance from the cloud top, although such dependence is not taken into account explicitly in most studies or treatments
in climate models. We have clarified such problems and proposed a practical solution for them in the present paper.


**5 Summary**
In the development of the climate model MRI-ESM2 that is planned for use in CMIP6 and CFMIP-3 simulations, the
representations of clouds are significantly improved from the previous version MRI-CGCM3 used in CMIP5 and CFMIP-2
simulations. The score of the spatial pattern of radiative fluxes at the top of the atmosphere for MRI-ESM2 is better than any
of the 48 CMIP5 models. In this paper, we presented comprehensively various modifications related to clouds, which
contribute to the improved cloud representation, and their main impacts. The modifications cover various schemes and
processes including the cloud scheme, turbulence scheme, cloud microphysics processes, the interaction between cloud and
convection schemes, resolution issues, cloud radiation processes, the aerosol properties, and numerics. Note that the
improvement of performance in climate models due to an update is ordinarily contributed by the cumulative effect of many
minor modifications rather than by the introduction of a new advanced scheme. In addition, the new stratocumulus
parameterization and improved cloud ice fall scheme are described in detail, because they include scientifically new concepts
and technically important issues. As a result, this paper will be useful for model developers and users of our CMIP6 outputs,
especially those related to clouds.
The most remarkable improvement addressed the serious lack of upward shortwave radiative flux over the Southern
Ocean in the old version. This improvement was obtained mainly by (i) an increase in low cloud cover due to the
implementation of the new stratocumulus scheme, a new treatment of the suppression of shallow convection under
stratocumulus conditions, and increased horizontal resolution for the radiation calculation, (ii) an increase in the ratio of
super-cooled liquid water due to the modified treatment of the WBF effect, and (iii) an increase in cloud droplet number
concentration by taking the effect of small size sea-salt aerosols into account. Items (ii) and (iii) contribute to an increase in
the optical thickness of clouds. The excessive reflection of solar radiation over the tropics in MRI-CGCM3 was substantially
reduced by the introduction of a new cloud overlap scheme, PICA. Increased vertical resolution from L48 to L80 and a
treatment of the suppression of shallow convection under stratocumulus conditions contribute to improve the vertical
structure of the transition from subtropical stratocumulus to cumulus. In addition, improved treatments of cloud ice

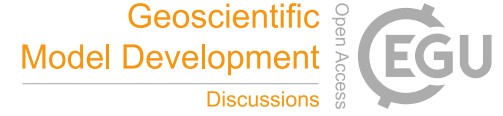

sedimentation and conversion of cloud ice to snow, which are based on more accurate physics than the old ones, alleviated
the strong time-step dependency of IWC.
However, the modifications in MRI-ESM2 are still relatively simple and ad hoc in some cases. Therefore, we should
continue to develop various schemes and processes related to clouds, especially cloud microphysics and the treatment of
cloud inhomogeneity within a model grid box, by introducing more sophisticated concepts.
On a final note, we acknowledge the many evaluation and intercomparison studies related to clouds for CMIP multi-
models, which have given us useful information for model development (e.g., Jiang et al. (2012) for vertical profiles of cloud
water content and water vapour; Su et al. (2013) for vertical profiles of cloud fraction and cloud water content under
different large-scale environments; McCoy et al. (2015) and Cesana et al. (2015) for ratios of super-cooled liquid water and
ice; Nam et al. (2012) for cloud radiative effect and vertical structure of low clouds; Nuijens et al. (2015) for vertical
structures and temporal variations of trade-wind cumulus; Bodas-Salcedo et al. (2014) for cloud and radiation biases over the
Southern Ocean; Kawai et al. (2018) for marine fog; Suzuki et al. (2015) for warm rain formation process; Tsushima et al.
(2013) for occurrence frequency and cloud radiative effect of each cloud regime). It is impossible for a modeller to examine
all of these characteristics in their own model, because there are many aspects to examine even for cloud related values alone
and these evaluations need specific knowledge and careful treatment. Therefore, these evaluation activities are very helpful
for modellers to improve and develop their models.


**Code and Data availability**
Access to the simulation data can be granted upon request. The MRI-ESM2 code is the property of MRI/JMA and not
available to the general public. Access to the code can be granted upon request, under a collaborative framework between
MRI and related institutes or universities. The code can be provided to the editor and the reviewers for the purpose of the
review of the manuscript.


**Appendix**
**A. Derivation of the conversion rate of cloud ice crystals to snow**
The conversion rate of cloud ice crystals to snow (cloud ice particles whose size is larger than 100 μm are called "snow"
here) in the new treatment is derived under the simple assumptions described below. Although these assumptions are rather
rough, the advantage is that this rate utilized in the scheme is derived from observational relationships for tropical cirrus.

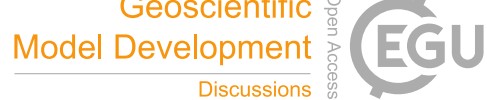



It is assumed that the ratio between cloud ice crystals and snow is not the same throughout a cloud, but depends on the
depth from the cloud top. It is presumed that the ratio of small cloud ice crystals is large near the cloud top and the ratio of
snow (large cloud ice) increases downward in the cloud, because upper cloud ice crystals are continuously converted to snow
and the density of snow, which falls with velocity much faster than cloud ice crystals, is accumulated downward. Therefore,
the ratios should be a function of the distance from the cloud top, and the ratios $\alpha_i$ in MH97 should be regarded as the ratio at
a certain distance from the cloud top.
To derive the conversion rate in this study, cloud ice content $q_i$ (kg kg$^{-1}$) was assumed to be vertically homogeneous in
the cloud. The snow density (kg m$^{-3}$) that is produced by a unit volume of cloud ice crystals existing at upper altitude is $C_{\mathrm{I2S}}$
$\rho_a v_s^{-1}$, using a conversion rate of cloud ice to snow $C_{\mathrm{I2S}}$ (kg kg$^{-1}$ s$^{-1}$). Consequently, the snow density at height $z$ can be
written as follows, using the cloud top height $z_{\mathrm{ctop}}$.
$$\int_{z}^{z_{\mathrm{ctop}}} C_{\mathrm{I2S}} \frac{\rho_a}{v_s} dz \approx \frac{z_{\mathrm{ctop}} - z}{v_s} \rho_a C_{\mathrm{I2S}}$$

where a constant value is used for $\rho_a$ regardless of the height for simplicity. Then snow content per unit air mass is $C_{\mathrm{I2S}} H_c$
$v_s^{-1}$ (kg kg$^{-1}$) using $H_c \equiv z_{\mathrm{ctop}} - z$. On the other hand, the ratio of cloud ice crystals to snow can be written as follows using the
observational function $\alpha_i$ by MH97:
$$q_i : \frac{H_c}{v_s} C_{\mathrm{I2S}} = \alpha_i : 1 - \alpha_i$$

Therefore, $C_{\mathrm{I2S}}$ can be derived as follows:
$$C_{\mathrm{I2S}} = \frac{1 - \alpha_i}{\alpha_i} \frac{v_s}{H_c} q_i$$


**Author contribution**
HK was responsible for most aspects of model developments related to the representation of clouds. SY performed
tuning of clouds simulated in MRI-ESM2 and many sensitivity tests. TK performed coding related to aerosol optical
properties and the output format of the model. NO and TT developed the aerosol model and contributed to the improvements
of the aerosol radiation and aerosol cloud interactions. RN developed PICA and HY implemented the scheme into MRI-
ESM2. SY and TK performed many model simulations, and TK and HY contributed to find coding problems in the original
cloud scheme. All authors contributed to related discussions. HK wrote the first draft of the article, and all authors
contributed to the writing of the final version of the article.



## Acknowledgements

Several datasets (Pincus et al., 2012; Zhang et al., 2012) used in this work were obtained from the obs4MIPs project (https://www.earthsystemcog.org/projects/obs4mips/) hosted on the Earth System Grid Federation (http://esgf.llnl.gov). This study was partly supported by "Integrated Research Program for Advancing Climate Models: TOUGOU" from the Ministry of Education, Culture, Sports, Science and Technology (MEXT), Japan. Additionally, it was supported by the Japan Society for the Promotion of Science (JSPS) KAKENHI Grant Numbers JP26701004, JP16H01772, JP18H05292, and JP18H03363 and the Environment Research and Technology Development Fund (2-1703 and S-12) of the Environmental Restoration and Conservation Agency, Japan. Kohei Yoshida made efforts to determine vertical model levels in MRI-ESM2, and Eiki Shindo kindly supported HK to perform model experiments.

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



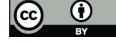


| Experiments | Section |
|---|---|
| Control   (time step = 3600 s, 1800 s [default], 900 s, and 300 s) | |
| with an old version of stratocumulus scheme | 3.1 |
| with an old treatment of the WBF effect | 3.2 |
| shallow convection can be active even under stratocumulus conditions | 3.3 |
| shallow convection can be active even under stratocumulus conditions using L48 | 3.4 |
| with an old version of cloud overlap scheme | 3.5 |
| radiation calculation for every two latitudinal grids | 3.6 |
| 1-hourly longwave radiation calculation | 3.6 |
| using original (not doubled) number concentration of sea salt CCN | 3.8 |
| with an old version of ice fall scheme   (time step = 3600 s, 1800 s, 900 s, and 300 s) | 3.9 |


**Table 1: List of sensitivity experiments performed in the present study using MRI-ESM2 to identify the effect of each modification.**
**The second column shows the section in which each modification is discussed.**






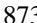



**Figure 1: (a, b) Climatologies of total cloud cover (%), (c, d) biases of total cloud cover (%) with respect to ISCCP observations, and (e, f) biases of upward shortwave radiative flux (W m⁻²) at the top of the atmosphere with respect to CERES-EBAF simulated by (a, c, e) MRI-CGCM3 and (b, d, f) MRI-ESM2. The climatologies cover the period 1986–2005 for model simulations and ISCCP observational data, and 2001–2010 for CERES-EBAF data.**











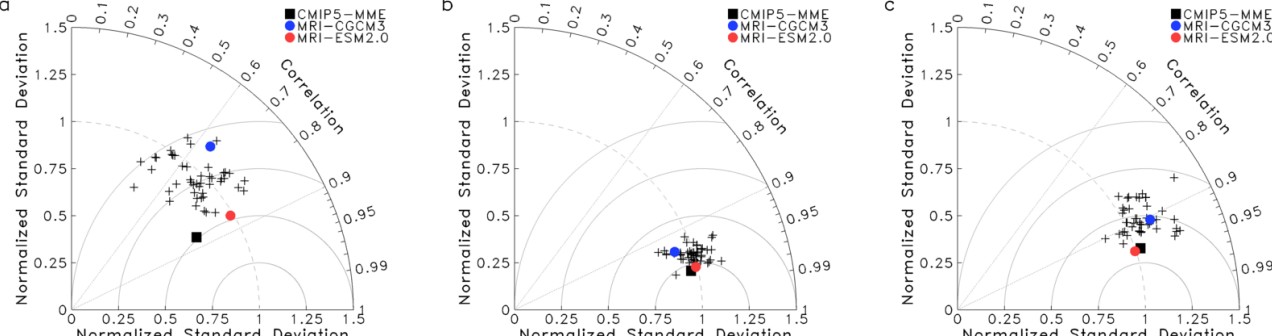


**Figure 2: Taylor diagrams for upward (a) shortwave, (b) longwave, and (c) net radiative fluxes at the top of the atmosphere for MRI-CGCM3 (blue dot), MRI-ESM2 (red dot), the CMIP5 multi-model mean (black square), and individual CMIP5 models (crosses). CERES-EBAF data are used as observations.**







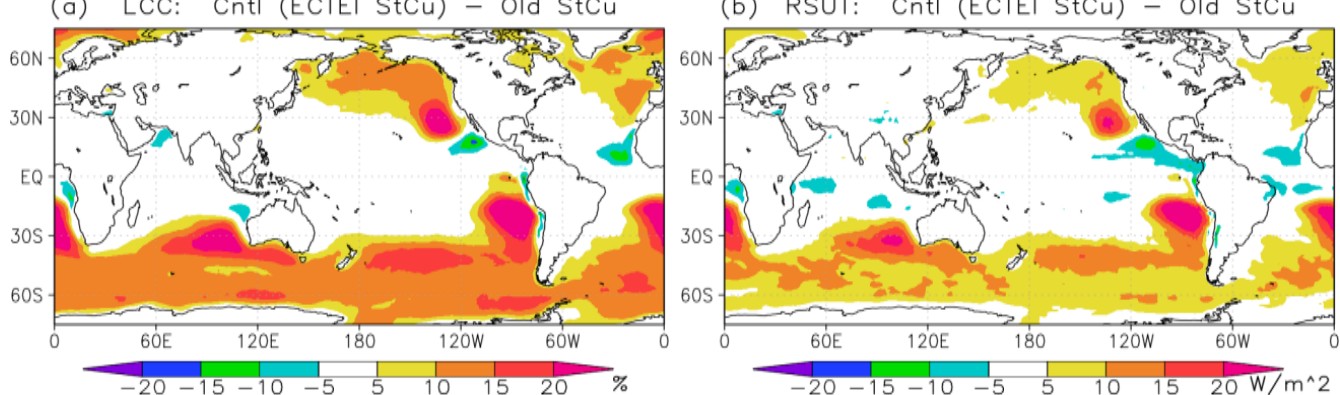


**Figure 3: Impacts of the new stratocumulus scheme on (a) low cloud cover (%) and (b) TOA upward shortwave radiative flux (W**
**m$^{-2}$). The plots show results for the control model (with the new stratocumulus scheme) minus those for an experiment with an old**
**version of the stratocumulus scheme.**







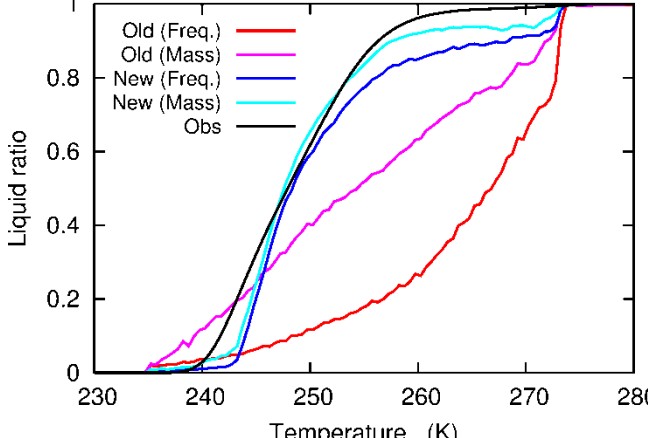



**Figure 4: Ratio of super-cooled liquid water to total cloud water as a function of temperature. The plot is obtained from snapshot global data for 10 days in July 2001 using the old (red and pink lines) and new (blue and light blue lines) treatments of the WBF effect. The ratios are calculated using two methods: mass weighted ratio (pink and light blue lines) in which liquid and ice masses are averaged over temperature bins first and the liquid water ratio is calculated from the averaged masses, and frequency ratio (red and blue lines) in which the snapshot ratio of liquid water is weighted by snapshot cloud fraction and averaged over temperature bins. Results from Hu et al. (2010) for observations are also shown (black line).**









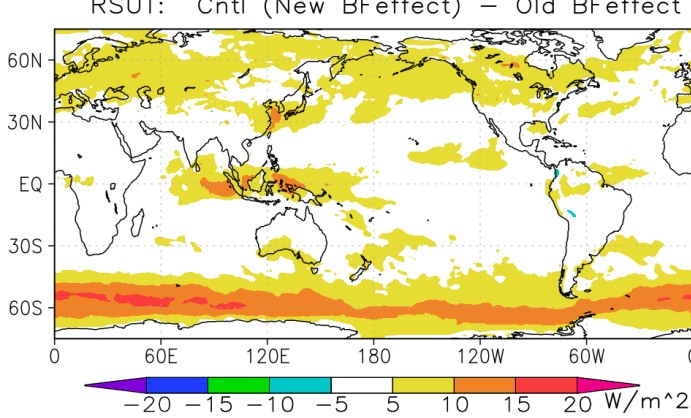


**Figure 5: Impact of the new treatment of the WBF effect on TOA upward shortwave radiative flux (W m$^{-2}$). The plot shows the results for the control model (with the new treatment) minus those for an experiment with an old version of the treatment.**









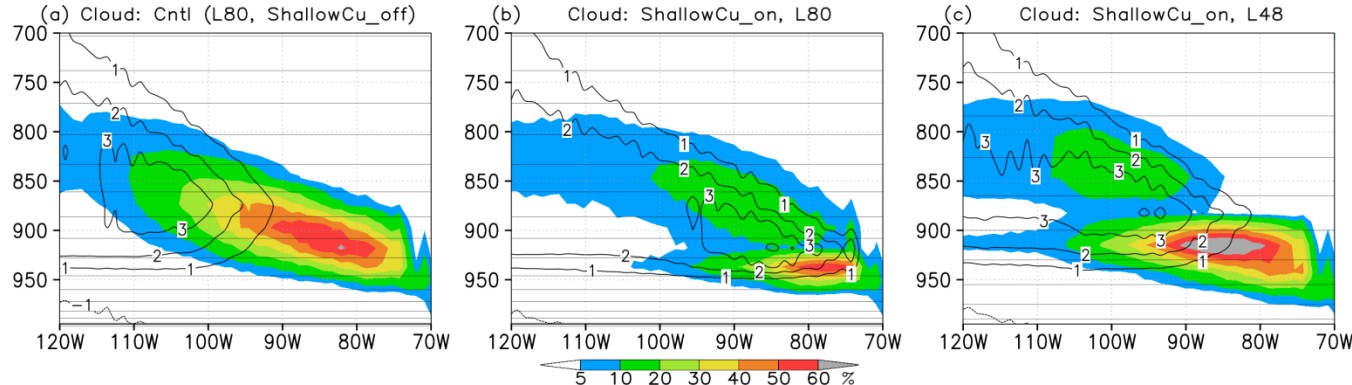



**Figure 6: Cross sections of cloud fraction (colour, %) along 20°S for January. (left) The control model (L80, a treatment of shallow convection suppressed under stratocumulus conditions), (middle) the same as the left panel but where shallow convection can be active even under stratocumulus conditions, and (right) the same as the middle panel except for vertical resolution L48. Horizontal straight lines show the vertical model layers and contours show the heating rate of the convection scheme (K day$^{-1}$).**




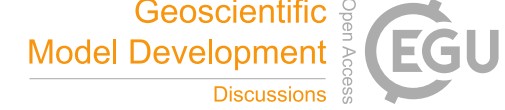




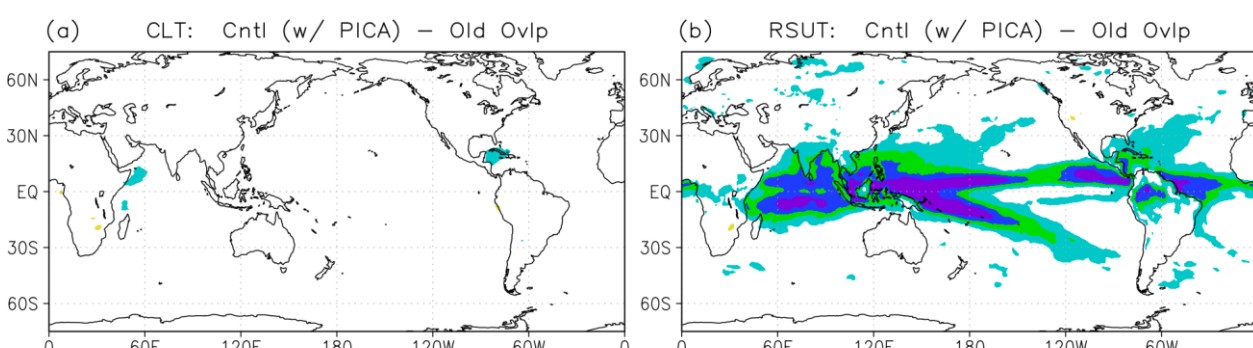



**Figure 7: Impacts of new cloud overlap scheme, PICA, for shortwave radiation calculation on (a) total cloud cover (%) and (b) TOA upward shortwave radiative flux (W m$^{-2}$). The plots show results for the control model (with PICA) minus those for an experiment with an old version of the cloud overlap scheme.**



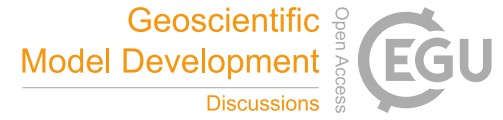




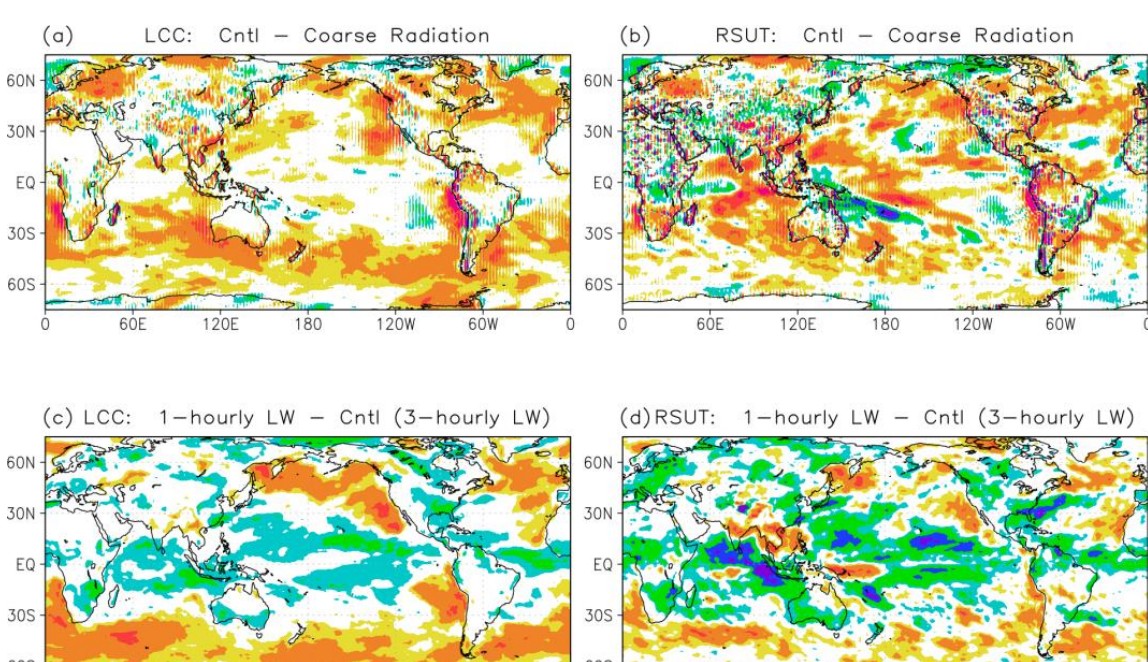



**Figure 8: Impacts of (a, b) increased horizontal resolution for the radiation calculation and (c, d) increased frequency of calculation for longwave radiation on (a, c) low cloud cover (%) and (b, d) TOA upward shortwave radiative flux (W m$^{-2}$). Panels (a, b) show results for the control model (calculation for every single grid box) minus those for an experiment with calculation for every two latitudinal grid boxes. Panels (c, d) show results for an experiment with 1-hourly longwave radiation calculation minus those for the control model (3-hourly calculation).**






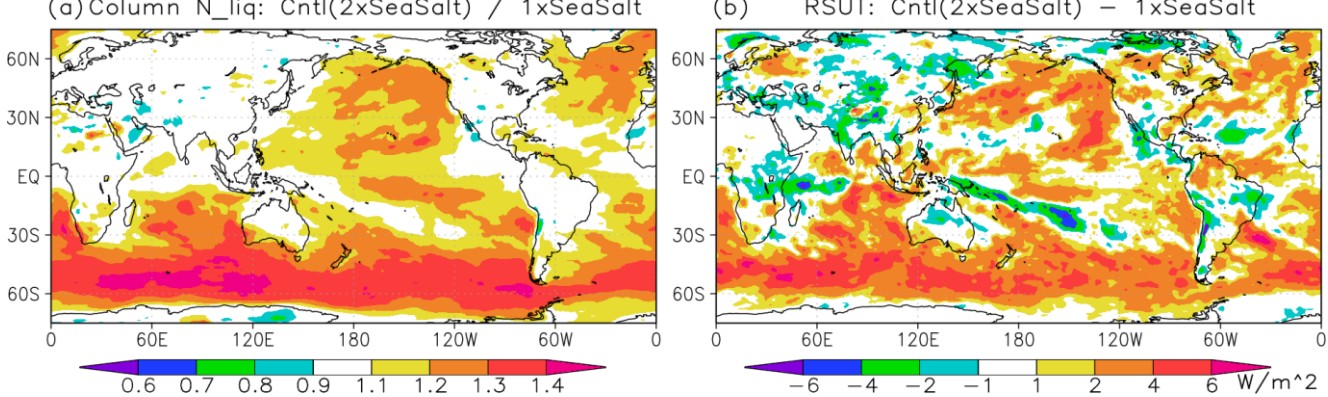



**Figure 9: Impacts of doubled number concentration of sea salt CCN on (a) column-integrated number concentration of cloud**
**droplets (unitless) and (b) TOA upward shortwave radiative flux (W m⁻²). The panels show the ratio (a) and the difference (b)**
**between results for the control model (doubled number concentration of sea salt CCN) and those for an experiment using the**
**original number concentration of sea salt CCN.**



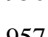



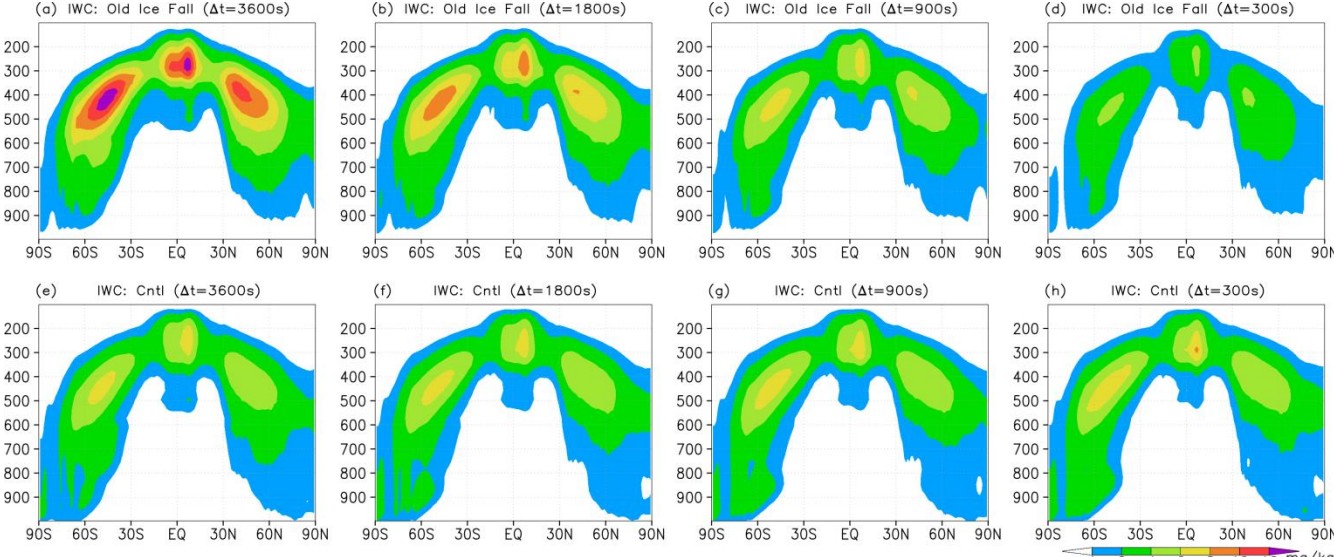



**Figure 10: Zonal average of ice water content (mg kg⁻¹) for different model time steps. Upper panels show results using the old ice**
**fall scheme and lower panels the control simulation using the modified ice fall scheme. From left to right, the time steps are 3600 s,**
**1800 s, 900 s and 300 s. The vertical axis shows air pressure (hPa) and the horizontal axis shows latitude.**




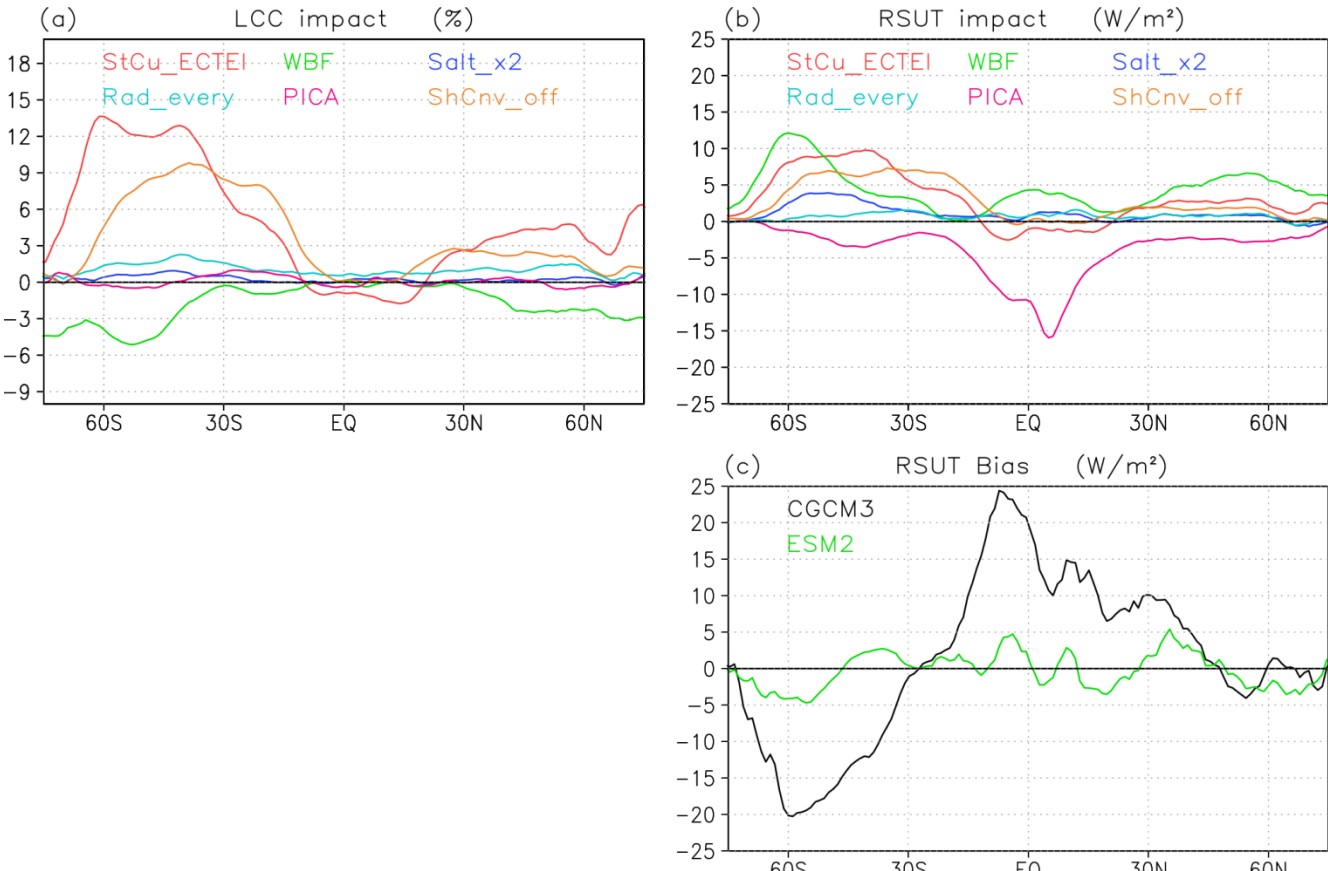

**Figure 11: Impacts of each modification on zonal means of (a) low cloud cover (%) and (b) TOA upward shortwave radiative flux (W m⁻²). Modifications include a new stratocumulus scheme (red line), the new treatment of the WBF effect (green), doubled number concentration of sea salt CCN (blue), increased horizontal resolution for radiation calculation (light blue), a new cloud overlap scheme, PICA (pink), and a treatment of shallow convection suppressed under stratocumulus conditions (orange). Each impact is calculated from the simulation data described in Section 2.3. The biases in TOA upward shortwave radiative flux for MRI-CGCM3 (black line) and MRI-ESM2 (green) are also shown in panel (c), where the data used are the same as in Fig. 1.**





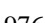


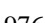

**Figure 12: Climatologies of low stratiform cloud cover (%), LTS (K), EIS (K), and ECTEI (K) for December to February (left panels) and June to August (right panels). Cloud cover data were obtained from EECRA shipboard observations and stability indexes were calculated using ERA-40 data (1957–2002).**





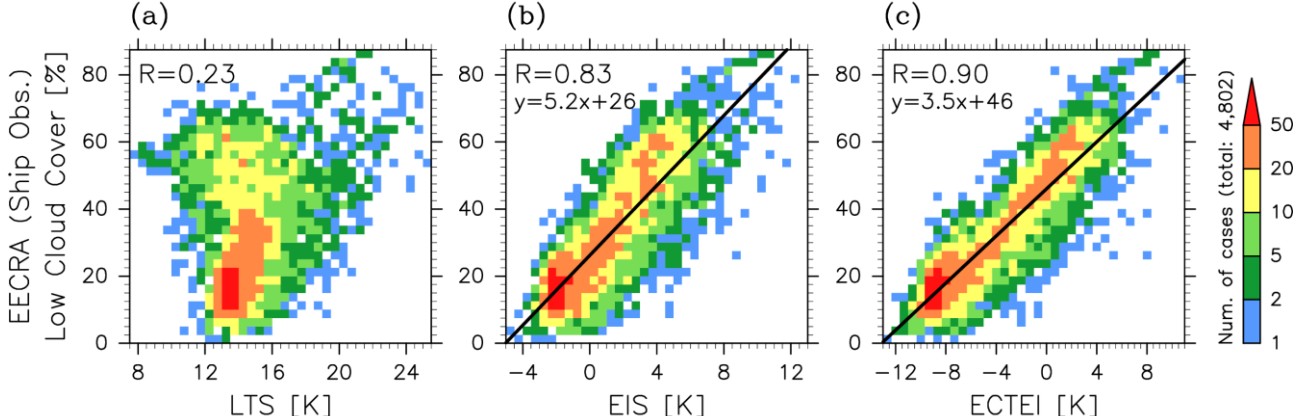

**Figure 13: Frequencies of occurrence of low stratiform cloud cover (combined cloud cover of stratocumulus, stratus, and sky-**
**obscuring fog) sorted by (a) LTS, (b) EIS, and (c) ECTEI ($\beta$ = 0.23), based on all 5° × 5° seasonal climatology data. Data are the**
**same as in Fig. 12 but all the data between 60°N and 60°S for all seasons were used. Linear regression lines and the correlation**
**coefficients are shown.**

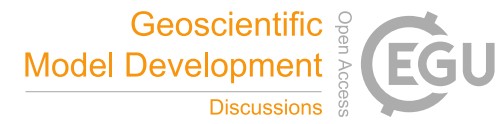


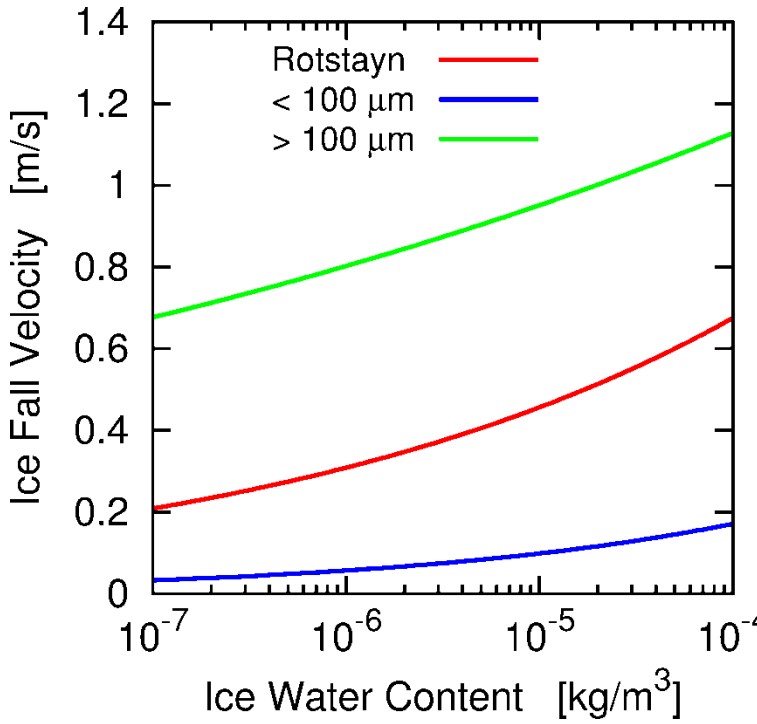

**Figure 14: Ice sedimentation velocities (m s$^{-1}$) of Rotstayn (1997) (Eq. (1), red line), derived for particles smaller than 100 µm (Eq.**
**(3), blue line), and for particles larger than 100 µm (Eq. (4), green line). The horizontal axis shows ice water mass density $\rho_a\, q_i$**
**(kg m$^{-3}$).**