# Peer review of "Significant Improvement of Cloud Representation in Global Climate Model MRI-ESM2"

_Geoscientific Model Development, 2019_

## Referee Comment (RC1) · Anonymous Referee #1 · 17 Apr 2019

This manuscript present how the cloud representation has been improved between two versions of the MRI climate model. This topic is very important because the representation of clouds in climate models is essential for both current climate and climate change simulations. The manuscript describes the origins of the defaults of previous model version and why and how the new developments allow to solve them in the new version. The manuscript is very well written, with a clear and complete presentation of the many development steps. It will be very useful for both people who analyses the results of this model and people who develops other climate or weather forecast models. In addition to the descriptive aspect, this manuscript contains very useful information to understand the physics of the phenomena, the hypotheses made, the possible numerical difficulties, etc.

[Figure]

The only point that could be better addressed is the link between the developments presented in the manuscript and the tuning of the whole climate model, i.e. when the atmospheric model is coupled with all the other components. As mentioned by the authors, the tuning is crucial and often overlooked. The tuning of some parameters is mentioned in the manuscript, but not how they have been tuned, with which target. In addition the tuning can be done in different successive steps, for instance when developing the parameterization and performing dedicated simulations, and/or in a later step when adjusting the whole climate model. Has this approach been used here? If yes, the values that are given correspond to intermediate values or values used within the full coupled model? When presenting and giving the value of the various parameters of the model, it would be very useful to specify (i) if the authors consider that this parameter can or should be adjusted or not, (ii) if it is the case what is the range of possible values, (iii) if a multi-step approach is used for the tuning, if the given values correspond to an intermediate step or to the final step, with the full coupled model.

I strongly recommend the publication of this article after including the minor improvement suggested above. I would like to thank the authors for the quality of their interesting manuscript, which is agreeable to read and easy to review.

Minor comment: please specify the unit of the variables in section 4.2.1

---

## Referee Comment (RC2) · Anonymous Referee #2 · 26 Apr 2019

This study by Kawai et al. takes an excellent approach to documenting model development and fits very well into GMD. My only very major concern is the excessive use of empirical relationships that arise as a consequence of a large number of physical processes in the stratocumulus and also the phase partitioning parameterizations in MRI-ESM2. While the underlying physics remain the same in a future climate, there is absolutely no guarantee that these empirical relationships can be applied to climate change (as detailed below). In my opinion, this point clearly deserves some discussion. In particular, there are indications that the empirical relationship between LTS (or EIS) and cloud cover that explains day-to-day variations does not hold in a climate change context (see below). Although day-to-day variations will still be governed by LTS in a warmer climate, the mean stratocumulus cloud cover might change in ways that have

nothing to do with this relationship (just think of two parallel and shifted regression lines perhaps with similar slopes, one of which represents the relationship in the present climate and the other the relationship in the future climate). While on a day-to-day basis larger LTS is associated with higher stratocumulus cloud cover, many climate models suggest that in a warmer climate, LTS increases while cloud cover in the stratocumulus regions decreases. Here, instead of the processes that are responsible for creating the day-to-day relationship, the actual present-day empirical relationship between EIS and cloud cover is baked into MRI-ESM2. This may cause a spurious negative contribution to the cloud feedback in future climate simulations, even if the relationship is only used to determine a threshold in the turbulence scheme. While using empirical relationships can serve to make results look good, using them is clearly not an ideal approach when constructing a climate model. Thus, in a climate change context, one should not only evaluate the performance of a model by comparing it to observations, but also take into account to what extent the parameterizations are based on simple physics and to what extent they are based on empirical relationships that arise from a combination of many processes. A climate model with imperfections in the representation of the current climate might in the end be preferable to a model that uses many empirical relationships to constrain the results. Using empirical relationships such as the relationship between LTS (or EIS) and cloud cover can lead to spurious climate feedbacks.

On the whole, however, I very much enjoyed reading this manuscript because of the systematic approach taken to evaluate the effect of changing the individual parameterizations. I think this can clearly serve as a good example for other groups. Probably most groups take an iterative approach to model development. In this process, often each step is evaluated separately, sometimes several changes are combined, and sometimes changes are reverted if they do not yield the desired effect. Ultimately, the successful groups certainly also form a fairly clear picture of the role of the individual changes. However, until recently a larger fraction of the resulting insights tended to be hidden in a few peoples heads. Kawai et al. use their new model version as a base model and then revert each development step, always starting from the same base

model. To me this seems a very nice and perhaps also novel way to make the steps more transparent to others. I think this is very useful. Rather than with just tuning (as in Mauritsen et al.), this manuscript deals with actual model development (although I do see some caveats as outlined above).

Perhaps, in a follow-up study the authors could take a similarly systematic approach to study the effect of each change on climate sensitivity and the aerosol.

Specific Comments:

1. The partitioning of detrained cloud water into liquid and ice in MRI-ESM2 follows Hu et al. (2010). The same data by Hu et al. (2010) was also used to evaluate MRI-ESM2. It is found that the mass and the frequency ratio in Fig. 4 are both fairly close to the Hu et al. data. Thus overall, the WBF process appears to have a rather small influence (which might be partially explained by still using a mixing ratio threshold). Could you please briefly comment on this? Is the WBF process allowed to act on detrained cloud water in the same time step?

2. Hu et al. (2010) present an empirical relationship that arises as a the consequence of a large number of physical processes. While the underlying physics remain the same, there is absolutely no guarantee that such a relationship will hold in a future climate. Updrafts and/or aerosol may influence such a relationship and they may change in a systematic way.

3. Many climate models show a decrease of cloud cover at the same time as LTS increases in a future climate. In a regional model, Lauer et. al 2010 (https://www.doi.org/10.1175/2010JCLI3666.1) also find that changes associated with global warming do not follow empirical relationships between LTS and cloud cover. Therefore, such empirical relationships should not be used in climate models.

4. I know that this would mean breaking a tradition, but I would be very interested in seeing a plot that shows the effect of the bug fixes mentioned in 3.7.

5. Do the the changes due to the individual model modifications add up linearly to yield the final result? In Fig. 11, a plot could be added that compares the sum of the individual contributions to the difference between MRI-ESM2 and CGCM3.

6. Fig. 10 shows a strongly decreased time step dependency of the ice water content in MRI-ESM2 compared to MRI-CGCM3 which the authors attribute to the I2S conversion term. This seems very plausible to me. However, if I understand it right, in MRI-ESM2, the partitioning of IWC into IWC>100 and IWC<100 is still performed at every time step (according to Eq. S10), right? As far as I can see, in the absence of a threshold below which conversion from cloud ice to snow starts to be active, this could in principle still cause a problem similar to the one described in lines line 423f. Nevertheless, at least the time step dependence seems to be addressed. Without first looking at Fig. 10, I would have expected that in the case of a long-lived non-precipitating cloud the ice water path would still become depleted during a successive time steps because of this partitioning. And, on the other hand, does partitioning every time step mean that a part of the IWP will always be be particles smaller than 100 micrometer (IWC<100)? Again, all of this seems less of a problem than what has already been addressed here.

p. 2, l. 46: ratio of supercooled water -> ratio of supercooled water to cloud (liquid and ice) water

p. 9, l. 281f: I completely agree on this (https://doi.org/10.5194/gmd-2018-307-RC1).

The advection term in Eqs. 2 and 6 is in flux form and not in advection form. In order to convert an advection equation to flux form, it is necessary to use the continuity equation. This implies that the flux form equation is valid in cases in which the tracer (here: ice) is advected with the flow. The flux from equation is not valid for sedimentation.

p. 13, l. 396f: Morrison and Gettelman, 2007 (https://www.doi.org/10.1175/2008JCLI2105.1) use substepping.

p. 15, l. 445f: this is not unavoidable. see my comment regarding l. 396f.

Fig. 4, caption: should the Hu et al. data be compared to the mass or the frequency weighted ratio?

---

## Author Comment (AC1) · 6 Jun 2019

Reply to Reviewer #1

We would like to thank the reviewer for the supportive and helpful comments to improve our manuscript. Below we address the reviewer's comments in detail.

**Comments from the Reviewer**

**- The reviewer suggested us to include some descriptions about the link between the developments presented in the manuscript and the tuning of the whole climate model**

We added a new subsection that briefly describes tuning issues as follows.
* * *
3.11 Comments on tuning

   At the end of this section, we give a brief description of the model tuning related to clouds. At a stage of developing schemes, a number of amip type simulations (with typical one-year length) were performed using atmospheric and aerosol coupled model, to check the basic behavior of schemes and the basic impacts on radiative fluxes. At a tuning stage, five-year runs of amip type simulations were mainly examined. The main targets for tuning parameters related to clouds in MRI-ESM2 were global-mean biases and root-mean square errors of shortwave and longwave radiative fluxes at the top of the atmosphere. The tuning parameters related to clouds are parameters which affect differently by cloud types and control cloud properties such as cloud cover, cloud water content, and cloud number concentration. In the stratocumulus parameterization (Section 3.1), the threshold value of ECTEI was tuned to increase Southern Ocean clouds as described in Section 4.1.3. The relatively large mode radius of sulfate of 0.10 µm (possible range: 0.05 − 0.10 µm) was chosen to obtain smaller cloud droplet number concentration to prevent an excessive aerosol-cloud interaction. Treatment of the WBF effect (Section 3.2), cloud overlap scheme (Section 3.5), schemes for ice sedimentation and ice conversion to snow (Section 3.9), and others (Sections 3.3, 3.4, 3.6, and 3.7) were not tuned. Descriptions of the model tuning (other than cloud-related parameters) are given in Yukimoto et al. (2019).
* * *
In addition, the possible range of ECTEI (−3.0 − +3.0 K) was specified in the first

paragraph in Section 4.1.3.

**- Minor comment: please specify the unit of the variables in section 4.2.1**

It is true that the unit in the section is important information for readers, as the reviewer pointed out. Thank you. The unit was specified.

---

## Author Comment (AC2) · 6 Jun 2019

We would like to thank the reviewer for the supportive comments and helpful suggestions to improve our manuscript. In our new version of the manuscript we try to follow virtually all of the reviewer's suggestions. Below we address the reviewer's comments in detail.

**Comments from the Reviewer**

**- there is absolutely no guarantee that these empirical relationships can be applied to climate change. In my opinion, this point clearly deserves some discussion.**

We added a new section and discuss this issue briefly as follows:
* * *
4.1.4 Brief discussion on climate change simulations

It is well-known that changes in LCC in warmer climates cannot be explained by changes in LTS (e.g., Williams et al., 2006; Medeiros et al., 2008; Lauer et al., 2010). The mechanism of this discrepancy is also well-understood; inevitable decrease of moist adiabatic lapse rate in the free atmosphere in warmer climates causes increase in LTS (e.g., Miller, 1997; Larson et al., 1999), even though the inversion strength that probably contributes to determine LCC does not change (e.g., Wood and Bretherton, 2006; Caldwell and Bretherton, 2009). It was expected that an index EIS could avoid this problem and could be used for discussion of LCC changes under warmer climates because EIS is a more physics-based index that represents inversion strength at the cloud top more directly. However, more recently, it turned out that LCC tends to decrease, although EIS increases in warmer climates in most climate models (e.g., Webb et al., 2013). Subsequently, it was shown by Qu et al. (2014) that changes (including variations in the present climate and future changes) in LCC can be determined by a linear combination of changes in EIS (positive correlation) and SST (negative correlation). Kawai et al. (2017) derived the linear combination from the index ECTEI and showed that a decrease in LCC under increased EIS in warmer climates can be explained based on the ECTEI change (see Kawai et al. (2017) for more detail). It is true that a usage of empirical relationships obtained in the present climate for climate change simulations has a possibility of causing spurious climate feedback. On the other

hand, we would like to note that ECTEI is even more physics-based index than EIS, the relationship is not used directly for cloud formation but used as a threshold for cloud top mixing, and ECTEI can explain positive low cloud feedback, although the risk of spurious climate feedback still cannot be eliminated.
* * *
- 1. The partitioning of detrained cloud water into liquid and ice in MRI-ESM2 follows Hu et al. (2010). The same data by Hu et al. (2010) was also used to evaluate MRI-ESM2. It is found that the mass and the frequency ratio in Fig. 4 are both fairly close to the Hu et al. data. Thus overall, the WBF process appears to have a rather small influence (which might be partially explained by still using a mixing ratio threshold). Could you please briefly comment on this? Is the WBF process allowed to act on detrained cloud water in the same time step?

We understand that this could be a little confusing part. First, source terms (to which Hu et al. (2010) function is adopted) are not only detrainment from convection, but also formation of stratiform clouds due to upward motion and temperature decrease. Actually, in MRI-ESM2 (with the new parameterization), production of ice from these source terms are dominant, and contributions from depositional growth (and others including immersion freezing, condensation freezing, and contact freezing) are much smaller. Therefore, the ratio in Fig. 4 is close to the Hu et al. function that is used to divide source terms into ice and liquid. This means that WBF process that explicitly occurs in MRI-ESM2 is very weak. (However, a usage of Hu et al. (2010) function to determine the ratio of newly produced LWC promotes ice production in the case of IWC greater than a threshold in our treatment, and it means WBF process is parameterized in MRI-ESM2.) The following short sentence was simply inserted in the third paragraph in Section 3.2: "In MRI-ESM2, IWC production from the source terms of LWC based on partitioning using a function of Hu et al. (2010) is dominant, and the contributions from a depositional growth and other freezing processes are considerably small."

- 2. Hu et al. (2010) present an empirical relationship that arises as a the consequence of a large number of physical processes. While the underlying physics remain the same, there is absolutely no guarantee that such a relationship will hold in a future climate. Updrafts and/or aerosol may influence such a relationship and they may change in a

systematic way.

The point is true. We added the following sentence in the last paragraph in Section 3.2: "It should also be noted that empirical relationships including the ratio curve of Hu et al. (2010) may not hold completely in a future climate because a large number of meteorological factors contribute to form such relationships and they may change in a systematic way."

- 3. **Many climate models show a decrease of cloud cover at the same time as LTS increases in a future climate. In a regional model, Lauer et. al 2010 (https://www.doi.org/10.1175/2010JCLI3666.1) also find that changes associated with global warming do not follow empirical relationships between LTS and cloud cover. Therefore, such empirical relationships should not be used in climate models.**

The response to this comment is included in the response to the reviewer's major comment.

- 4. **I know that this would mean breaking a tradition, but I would be very interested in seeing a plot that shows the effect of the bug fixes mentioned in 3.7.**

It is not easy to take back all the bugs and to convince our colleagues to spend computer resources to investigate influences of the bugs. Instead, the impact of the bug related to number concentrations of the cloud particles that is mentioned in Section 3.7 is shown in Fig. R1 (attached to this reply) for the reviewer (and the readers of this open discussion). The figure shows that the bug caused excessive reflection of solar radiation, particularly for stratocumulus and stratus over the subtropics and northern Pacific region for July.

- 5. **Do the the changes due to the individual model modifications add up linearly to yield the final result? In Fig. 11, a plot could be added that compares the sum of the individual contributions to the difference between MRI-ESM2 and CGCM3.**

We agree that it is an interesting issue. Actually, the sum of impacts from the individual model modifications described in this manuscript (in Fig. 11b) does not match the difference between MRI-ESM2 and CGCM3 (in Fig. 11c) so well. There are several

reasons for that. First, several bug fixes related to clouds significantly (as describes in the response above) contribute to the radiation flux bias. Second, modifications in some other physical processes, for example, convection parameterization contribute the radiation flux as well. Modifications in other component models, for example, the aerosol model (through changes in optical thickness of clear sky and optical thickness of clouds via changes in aerosol concentrations) contribute the radiation bias. In addition, there should be non-linearly on each impact from each modification. Therefore, we decided not to discuss this summation to avoid confusion and complexity.

- 6. Fig. 10 shows a strongly decreased time step dependency of the ice water content in MRI-ESM2 compared to MRI-CGCM3 which the authors attribute to the I2S conversion term. This seems very plausible to me. However, if I understand it right, in MRI-ESM2, the partitioning of IWC into IWC>100 and IWC<100 is still performed at every time step (according to Eq. S10), right? As far as I can see, in the absence of a threshold below which conversion from cloud ice to snow starts to be active, this could in principle still cause a problem similar to the one described in lines line 423f. Nevertheless, at least the time step dependence seems to be addressed. Without first looking at Fig. 10, I would have expected that in the case of a long-lived non-precipitating cloud the ice water path would still become depleted during a successive time steps because of this partitioning. And, on the other hand, does partitioning every time step mean that a part of the IWP will always be be particles smaller than 100 micrometer (IWC<100)? Again, all of this seems less of a problem than what has already been addressed here.

The situation is a little complicated.

First, it is important to confirm that the ratio of Eq. (S10) is not used to actually (directly) determine the ratio of large ice and small ice in total IWC. Therefore, large ice production during one time step (= $C_{I2S}\Delta t$) (not $(1-\alpha_i)\, q_i$) is basically proportional to $\Delta t$. (Note also that a ratio $r_{iw}$ is in Eq. (2) but a ratio $\alpha_i$ is not in Eq. (6). If, for example, the ratio $\alpha_i$ is used as a substitute for $r_{iw}$ in Eq. (2), similar timestep dependency is found.)

It is true that the ratio $\alpha_i$ is calculated from Eq. (S10) every time step. But it is used only to determine the conversion rate of small ice to large ice $C_{I2S}$ (Eq. (5)). To determine the conversion rate, the ratio $\alpha_i$ calculated from Eq. (S10) based on observation (at the observation depth from the top of the cloud) should be used and the ratio calculated in the model that depends on model time step should not be used. For clarification, the following sentence was inserted after Eq. (6): "Note that although the ratio $\alpha_i$ obtained from Eq. (S10) is used to calculate the conversion rate $C_{I2S}$, it is not used to directly

determine the ratio between small ice crystals and snow differently from in Eq. (2)."

It is true that in the case of a long-lived non-precipitating cloud the ice water path would still become depleted if we don't introduce the conversion threshold. But it would occur regardless of the model time step.

**- p. 2, l. 46: ratio of supercooled water -> ratio of supercooled water to cloud (liquid and ice) water**

Corrected.

**- p. 9, l. 281f: I completely agree on this (https://doi.org/10.5194/gmd-2018-307-RC1).**

Thank you for your agreement. We are pleased to know that the reviewer recognizes the value of non-use of a lower limit of cloud droplet number concentration.

**- The advection term in Eqs. 2 and 6 is in flux form and not in advection form. In order to convert an advection equation to flux form, it is necessary to use the continuity equation. This implies that the flux form equation is valid in cases in which the tracer (here: ice) is advected with the flow. The flux from equation is not valid for sedimentation.**

The reviewer's comment is correct. The equations in the previous manuscript were not correct. (We wanted to make the equations easy to understand, though they were forms not actually used. But it is true that it is confusing.) The equations that we actually use are as follows:

$$\frac{\partial q_i}{\partial t} = C_g + \frac{R_i}{\rho_a \Delta z} - \frac{v_{\text{ice}}}{\Delta z} r_{iw} q_i - \frac{(1 - r_{iw}) q_i}{\Delta t} \tag{2}$$

$$\frac{\partial q_i}{\partial t} = C_g + \frac{R_i}{\rho_a \Delta z} - \frac{v_i}{\Delta z} q_i - D_{\text{I2S}} q_i \tag{6}$$

where $R_i$ (kg m$^{-2}$ s$^{-1}$) is the ice sedimentation flux into the layer from above, and $\Delta z$ (m) is the layer thickness. (This form is generally used for ice sedimentation calculation (e.g., Smith, 1990; Rotstayn, 1997).)

Therefore, we corrected these equations as noted above. Thank you for your insightful comment.

- p. 13, l. 396f: Morrison and Gettelman, 2007 (https://www.doi.org/10.1175/2008JCLI2105.1) use substepping.

Thank you for the information. The following sentence was inserted:
"Although adopting shorter time steps for selected processes that is called substepping (e.g., Morrison and Gettelman, 2008) would be an ideal solution, it can increase computational cost to some degree."

- p. 15, l. 445f: this is not unavoidable. see my comment regarding l. 396f.

The sentence was modified as follows:
"Therefore, the sedimentation cannot be calculated appropriately with the time step used in our climate models, and the treatment of instant fall of snow (large ice) through to the surface is unavoidable, unless substepping is introduced."

- Fig. 4, caption: should the Hu et al. data be compared to the mass or the frequency weighted ratio?

The part in the caption was clarified as follows:
"An observational curve from Hu et al. (2010) that corresponds to a frequency ratio"

[Figure]

**Figure R1.** Impacts of a bug fix (bug fixed – with the bug) on TOA upward shortwave radiative flux (W m$^{-2}$) in amip type simulations for January (left panel) and July (right panel) in 2001.